# Radiomics-Based Analysis in the Prediction of Occult Lymph Node Metastases in Patients with Oral Cancer: A Systematic Review

**DOI:** 10.3390/jcm12154958

**Published:** 2023-07-28

**Authors:** Serena Jiang, Luca Giovanni Locatello, Giandomenico Maggiore, Oreste Gallo

**Affiliations:** 1Department of Otorhinolaryngology, Careggi University Hospital, Largo Brambilla 3, 50134 Florence, Italy; 2Department of Otorhinolaryngology, University Hospital “Santa Maria Della Misericordia”, Azienda Sanitaria Universitaria Friuli Centrale (ASUFC), 33100 Udine, Italy; locatello.lucagiovanni@gmail.com

**Keywords:** radiomics, head and neck, cancer, oral squamous cell carcinoma, oral carcinoma artificial intelligence

## Abstract

Background: Tumor extension and metastatic cervical lymph nodes’ (LNs) number and dimensions are major prognostic factors in patients with oral squamous cell carcinoma (OSCC). Radiomics-based models are being integrated into clinical practice in the prediction of LN status prior to surgery in order to optimize the treatment, yet their value is still debated. Methods: A systematic review of the literature was conducted according to the PRISMA guideline. Baseline study characteristics, and methodological items were extracted and summarized. Results: A total of 10 retrospective studies were included into the present study, each of them exploiting a single imaging modality. Data from a cohort of 1489 patients were analyzed: the highest AUC value was 99.5%, ACC ranges from 68% to 97.5%, and sensibility and specificity were over 0.65 and 0.70, respectively. Conclusion: Radiomics may be a noninvasive tool to predict occult LN metastases (LNM) in OSCC patients prior to treatment; further prospective studies are warranted to create a reproducible and reliable method for the detection of LNM in OSCC.

## 1. Introduction

Oral squamous cell carcinoma (OSCC) is the eighth most common malignancy worldwide [1]. It has a poor prognosis, with an overall 5-year survival rate of around 45–55% depending on the series considered [2]. This figure is much lower (around 20–30% at five years) especially in advanced stages according to the eighth edition of the AJCC/UICC [3]. The major prognostic factors are depth of invasion (DOI) > 5 mm, extranodal extension, positive or close surgical margins, pT3 or pT4 tumor (i.e., larger than 4 cm or infiltrating bony structures such as the mandible), pN2 or pN3 nodal disease, perineural invasion, vascular invasion, and lymphatic invasion. In particular, the presence of lymph node metastases (LNM) alone is known to reduce survival by approximately 50% [4].

The standard of care for OSCC is complete surgical resection with sufficient surgical margins (at least 5 mm are deemed necessary), followed by adjuvant radio-/chemotherapy in properly selected cases where the aforementioned adverse prognostic features are present. The following therapeutic strategies are currently available in managing a clinically negative (cN0) neck in early stage OSCCs [5]:Elective neck dissection (ND): which is associated with esthetic and functional morbidity and it represents a procedure that may affect negatively the quality of life of the patient; the decision on whether to perform or not ND in all cases of cN0 neck is still under debate [6];Watch and wait policy: this is currently disregarded as a valid option because it was substantially demonstrated that elective neck dissection resulted in longer overall and disease-free survival than did therapeutic neck dissection after nodal relapse [7];Sentinel node biopsy (SNB): in 2015, the Sentinel European Node Trial (SENT) reported an overall sensitivity and negative predictive value of 86% and 95%, respectively [8], and this strategy may be considered the current gold standard for early stage OSCC [9].

The introduction of artificial intelligence (AI) and its application to clinical decision-making in order to individualize patient care has become a major topic of discussion. Radiomics is a machine-learning (ML) approach for image analyses using advanced mathematical analysis [10].

In recent years, due to the development of ML algorithms coupled with more accessible digital data, more and more researchers have begun to focus on predicting molecular biomarkers, therapeutic responses, and survival prognostic factors in patients with head and neck (HN) carcinomas by extracting radiomics information features (e.g., shape description, intensity, or texture characteristics) from different imaging patterns (e.g., CT, MRI, PET, ultrasound images) [11]; in Mossinelli’s [12] retrospective study on 79 patients with oral tongue squamous cell carcinoma (OTSCC) MRI-based radiomics represents a promising noninvasive method of precision medicine, improving prognosis prediction before surgery.

Different non-invasive strategies exist for the prediction of LN status: clinical examination by digital palpation, neck imaging by ultrasound/CT/MRI potentially coupled with fine-needle aspiration of the suspected nodes, DNA microarray gene-expression profiling [13], nuclear medicine techniques such as positron emission tomography, the degree of differentiation of the primary tumor or the depth of invasion [14], but the gold standard is postoperative histopathological examination of the LNs. As a matter of fact, only a detailed (by simple microscopy and by techniques of immunohistochemistry) examination of the excised specimens can allow a surgical pathologist to identify micrometastases, which would have otherwise been overlooked. Despite the fact that ND is associated with many potential surgical complications, it remains true that up to 30% of early stage disease has occult cervical micrometastatic disease [15].

If we rely only upon preoperative standard imaging techniques, we know that lymph nodes larger than 10 mm are considered abnormal, yet around 20% of such nodes are pathologically free of disease, while up to 23% of nodes that show histological extracapsular spread measure less than 10 mm. Other features such as the presence of intranodal necrosis or irregular margins may indicate cancerous involvement but with variable accuracy [16,17].

In order to improve the diagnostic yield of these techniques, radiomics analyses have been successfully applied to predict the LN status of colorectal [18], cervical [19], and bladder cancer [20]. The role of radiomics in the assessment of occult lymph nodes in OSCCs patients have never been addressed to the best of our knowledge, and the aim of the present systematic review is to summarize the currently available clinical evidence on this topic while highlighting the unmet needs in this context.

## 2. Materials and Methods

### 2.1. Searching Strategy and Selection Criteria

Following the Preferred Reporting Items for Systematic Reviews and Meta-Analyses (PRISMA) guideline [21] we conducted a literature search of articles published from the beginning up to February 2023, using PubMed, Embase, Cochrane Library, and Scopus in order to identify the relevant studies. The following keywords were used: “radiomics AND oral cancer OR tumor”.

We included all original studies that implemented radiomics-based algorithms for analyzing preoperative imaging in patients with proven histology of OSCC. Articles were excluded based on the following criteria: studies with less than 10 patients or case reports, meeting abstracts, review/meta-analysis, and data not clearly stating the diagnostic performance.

The present systematic review is unregistered.

### 2.2. Data Collection

The title and abstract of the selected papers were carefully read according to the inclusion and exclusion criteria and duplicates were removed. We extracted data from each study, which were reviewed for consistency among the authors, and any discrepancies were resolved by consensus. The full text of the included studies was then read in order to extract the following data:

Reference: first author, year of publication, and country;

Study design (retrospective, prospective);

Preoperative imaging technique;

Where the predictive imaging features were extracted from (primary tumor, cervical lymph nodes);

Software used for the radiomics-based analysis;

Recruitment time span;

Sample size: divided into primary/train cohort and validation/test cohort;

Tumoral subsite of the oral cavity and staging (TNM 8th edition);

Number of positive and negative LNs or number of patients with positive and negative nodes;

Diagnostic quantitative data: sensitivity, specificity, accuracy (ACC), area under the receiving operator curve (AUC).

### 2.3. Definition of the Outcomes, Synthesis of the Literature, and Meta-Analysis

In manuscripts where multiple ML models were implemented, we have chosen the one with the highest AUC value. Due to the heterogeneity of the preoperative imaging techniques, the segmentation and features extraction, it was not possible to meta-analyze the papers; it was thus decided to critically discuss all the articles qualitatively.

### 2.4. Quality Assessment and Statistical Methods

The quality and the risk of bias of the articles included in this review were evaluated by the Quality In Prognosis Studies (QUIPS) tool with any discrepancies resolved by consensus by the authors [22]. Visualization of the risk-of-bias assessments was performed by creating a traffic lights plot using the robvis tool (version 0.3.0.900) [23].

## 3. Results

A flowchart of the study selection process is reported in Figure 1. We identified a total of 419 articles, we excluded 35 duplicates and 301 records because they were not relevant; out of the 63 papers screened, a total of 10 manuscripts were selected for in-depth analysis as shown in Table 1.

The majority of the articles (70%) were published in 2022, one in 2020, one in 2021 and one in 2019, while none of the included articles was published before 2019.

All the studies were retrospective in nature and most of them were based on single-center evaluation with a variable number of patients (total *n* = 1489; range = 40–313). The preoperative imaging study was made using MRI in five studies, CT in four, and PET in a remaining one.

A total of 60% of the articles focused on primary carcinoma of the tongue, amongst other oral cavity subsites (gingiva, floor of mouth), with a predictable spotlight on early stages (stage I–II).

In more than half of the cases, a validation cohort was screened using the same criteria as that for the primary cohort. Where the segmentation subsite was the tumor, the partition into the validation and the primary cohort was made among the patients; on the contrary, when the subsite was the LN, the division was made among the examined LNs.

In 60% of the included articles, the predictive features for occult LNM were derived from radiological features of the primary tumor, while in 30% they were derived from the features of the LNs; overall, the most accurate diagnostic models were derived using tumor-based features.

The diagnostic performances of the included studies are summarized in Table 2. Wang et al. [26] reported the highest AUC value (0.995), meanwhile, the least value was observed by Kudoh [30] (0.79); ACC ranges from 0.68 to 0.975; sensibility and specificity, when reported, are over 0.65 and 0.70, respectively, in two out of seven they were above 0.90.

Only three studies included a comparative analysis of the ML model with the radiologists’ evaluation: Committeri et al. [29] demonstrated a better performance of radiomics over the clinician’s performance, in Ren’s study [33] they were similar, and Wang Y [24] reported slightly worse values for ML model. Expectedly, the combination of clinical and ML models outperformed the single modality.

A traffic lights plot was created to visualize the risk-of-bias assessment (Figure 2), with a moderate-to-low risk of bias among all the included studies.

## 4. Discussion

In the present systematic review, we examined the use of radiomics-based analysis for the detection of occult neck metastasis in OSCC. Given the prognostic value of any nodal metastasis, early detection of OSCC and a comprehensive therapeutic strategy for both the primary tumor and the associated lymph nodes are of utmost importance.

Radiomics is a growing area of research that extracts and models medical image features using ML methods. Its goal is to implement AI algorithms in order to create a more accurate, cost-effective, and patient-tailored diagnostic and/or therapeutic tool. In the literature there are multiple studies that use radiomics for HN tumors: various authors critically reviewed the developments in diagnostic and therapeutic approaches in nasopharyngeal [34], laryngeal [35], thyroid [36], and salivary gland tumors [37]. Giannitto et al. [38] focused the attention on the diagnostic accuracy and methodological quality items in radiomics-based ML for the diagnosis of LNM in patients with HN cancer; however, the review did not discriminate outcomes based on tumor subsite, with almost half of the patients being affected by thyroid tumor. Additionally, Romeo et al. [39] used a similar approach in the prediction of tumor grade and nodal status on oropharyngeal and oral carcinomas.

ML models can be applied potentially to all imaging techniques, although it is preferentially used for more standardizable and reproducible ones, such as CT and MR. In the included papers, only Kudoh et al. [30] processed PET images and, interestingly, they reported the lowest diagnostic performance. Moreover, like the vast majority of studies of radiomics in HN, the studies herein analyzed are based on a single imaging modality.

Apart from the chosen imaging protocol, methodological heterogeneity is present also in the delineation of the region of interest (ROI), in the software used for image elaboration and radiomics feature extraction and processing.

Useful data on nodal status can be obtained even from the primary tumor mass because it is probably related to tumor biological heterogeneity and aggressiveness. As a matter of fact, the majority of the studies in our review extracted features from tumors (6), while only three were from LNs.

By focusing on these latter, in Wang’s article [24] the inclusion criteria were as follows: histopathologically proven OSCC who underwent ND and preoperative MRI contrast-enhanced scans of the head and neck. The LN with the “largest volume or with unclear edges and internal necrosis”, that is radiologically suspicious lymph nodes, were selected as the ROI. Only eight features were used to build the radiomics model. Tomita’s study [25] also included patients with histologically proven OSCC with benign or metastatic cervical LNs and available preoperative contrast-enhanced CT data. ROIs were drawn across all slices of the cross-sectional areas of the targeted LNs, that is those levels that were known to harbor micrometastases at final histopathology. For the evaluation of the CT scans, three radiologists independently assessed the LN status using specific criteria to determine if they were considered metastatic. The AUC values of the best ML-based model were superior to those of each individual human reader (*p* < 0.05); additionally, there were significant differences in specificity and diagnostic accuracy rates between them, demonstrating the potential of radiomics analysis in improving the accuracy of LN status assessment compared to human readers. Lastly, Kubo et al. [27] focused on cN0 patients diagnosed with tongue cancer who received treatment aimed at the primary tumor site without additional therapies (elective ND, chemotherapy). For patients that developed occult cervical LNM, but with no recurrence of the primary tumor, salvage surgery was performed, and histological analysis confirmed the presence of metastatic squamous cell carcinoma in these LNs. To analyze the CT scans, two radiation oncologists manually contoured each neck node level slice by slice in the axial plane rather than the primary tumor. It is crucial to point out that Wang Y [24] and Tomita [25] conducted an analysis on patients with suspicious LN that could be detected preoperatively on the radiological scans and then underwent ND. By comparing the results of AI-based analysis with clinical assessments, they can evaluate the potential of ML-based models as a tool in diagnosing and predicting malignancy in patients with positive LNs. Unsurprisingly, the choice of the ROI where to perform the extraction of the features was performed retrospectively. ML has the main objective to identify occult nodal disease: even if the exact ROI may be ambiguous where no radiologically suspicious region exists by definition, the anatomical levels most at risk (e.g., ipsilateral I-II-III levels for a cancer of the lateral tongue margin) remain those to assess with the focus on LNs rather than the primary tumor site, the researchers aim to improve the accuracy of cervical nodal staging; furthermore, the primary tumor itself could sometimes be challenging to contour accurately due to artifacts or its small size [27], while contouring neck nodes could be a more standardized process with fewer variations among different individuals.

Divergence in the qualitative items comes along with quantitative differences (e.g., number of images and/or features extracted). It is misleading to evaluate the performance of ML when diverse choices, selection methods, and classifiers are applied because the resulting models become sensitive to perturbation, contamination, and leakage of data.

Concerning the performance evaluation, and by excluding Kudoh’s results [30], AUC is over 82% in the reviewed studies. Only three out of ten conducted generalizability assessment with an independent (*n* = 1) or an external validation cohort.

There are many articles that introduce conventional imaging methods to predict cervical LN status early in OSCC. Van den Brekel et al. [40] compared the performance of ultrasound, CT, and MRI in 88 cN0 necks: sensitivity, specificity, and accuracy were for ultrasound 58%, 75%, and 68%, for CT 49%, 78%, and 66%, for MRI 55%, 88%, and 75%, respectively; FDG-PET studies reported sensitivity and specificity are quite variable: although this imaging modality is very useful in differentiating between benign and metastatic cervical LNs, inflammation and small nodal size can affect the nodal status assessment [41].

The current review found five studies that reported the traditional diagnostic performance of the radiologists; it is interesting to note that the average AUC, ACC, sensibility, and specificity curves of the clinicopathological factors were not always lower than those of the radiomics features. Wang Y et al. [24] found that the AUC of the model of MRI radiomic features was 0.88, which was better than that of the ADC and LN size; also Tomita [25] claimed that the radiomics approach yielded better diagnostic performance for differentiating between benign and metastatic cervical LNs than conventional CT; in Wang’s article [26] multivariate logistic regression analysis identified MRI-reported LN status (OR 2.432, 95% CI, 1.093–5.411) as an independent predictor of LNM. Kudoh [30] demonstrated that the 18F-FDG PET-based model had better potential for diagnosing cervical LNM and predicting late LNM in patients with OSCC than the clinicopathological factors model. Eventually, none of them performed a decision curve analysis to offer clinical guidelines for the preoperative management of the patient.

The specificity of ultrasound-guided FNAC is approximately 100%, advantages of the technique are its relatively low cost, lack of radiation exposure, and low-threshold availability; the main drawbacks are the sampling error of the aspirate due to the small size or inaccessibility of the LN and the operator-dependent nature of the procedure. If radiomic features of the primary tumor can outperform diagnostic assessment of the neck with imaging or ultrasound-guided FNAC is a demanding query to which we cannot give an answer yet [42]. However, cross-sectional imaging has the advantage to perform a full assessment of the lymph nodes, while FNAC is capable to sample only a part of it where the tumoral cells might not be identified.

The ambition of AI-based models is to help clinical evaluations in detecting occult LNM in OSCCs and unfortunately, a meta-analysis could not be conducted for the aforementioned methodological issues: this is the first limitation of our work.

Other limitations that must be acknowledged are the failure to validate model performance on a large, independent, external data set that prevents the applicability of findings to populations at large scale; the absence of well-structured, public/open, and worldwide “big data”, and of the methods used for training. AI-based algorithms notably require an enormous quantity of input information, therefore even in the face of over a thousand patients, we are far from reaching a definite answer in this field [43]. In this regard, the standardization of the automated methods, and the availability of high-quality open-source data seem imperative. Moreover, no prospective studies have been conducted and there is still the problem of “overfitting” which happens when AI gives undue importance to spurious correlations within past data.

All the reviewed articles are retrospective and they support and echo these findings.

This systematic review is poor in terms of clinical utility evaluation. We conducted a meticulous and independent search, according to PRISMA guidelines, of multiple online-available databases in order to provide an overview of the best performance of radiomics in LN status characterization in OSCCs; we wanted to highlight the strengths of this analysis but also the weak points, in order to create a shared approach in terms of both feature computation and methodology that will hopefully move this field of research to the routine clinical practice.

This is a rapidly evolving research area. Nowadays, we can talk about “multi-omics” data analysis (radiomics, genomics, proteomics, and metabolomics) that can be integrated with clinicopathological factors to help in accurate disease prediction, patient stratification, and delivery of precision medicine [44].

Radiomics prediction model has the potential to become a non-invasive diagnostic tool for HN cancer and LN status before treatment. By digitizing and analyzing the medical image data, the model’s predictions become more objective and standardized, thus reducing potential subjectivity in the diagnosis process and human error. Secondly, models can be validated and modified as more data becomes available, further enhancing its accuracy and reliability; moreover, AI may support inexperienced doctors in the assessment of lesions. This possibility also introduces medico-legal issues since the medical human judgment can fail as well as the AI: who would be liable if a mistake is made during AI-enhanced decision-making—such as ML-aided radiological diagnosis [45,46].

Cost-effective AI models can allow hospitals to incorporate the latter into daily clinical use; in order to make it happen, in addition to the development of a shared database of different medical centers from all over the world, prospective studies with a uniform and standardized imaging and processing protocol applied on a large and homogeneous cohort with an independent and/or external validation cohort should be conducted.

## 5. Conclusions

This systematic review provides an overview of the performance of radiomics-based models concerning the LN status in OSCCs. From our preliminary findings, the addition of AI-based models in the assessment of preoperative imaging may satisfactorily improve the detection of pathological lymph nodes in OSCC’s patients. Future reproduction of our results in other cohorts and by a uniform analytical protocol is anticipated. Finally, a proper clinical validation of these models in terms of oncological endpoints such as survival and disease-free recurrence is needed before incorporating these models in the decision-making process for these patients.

## Figures and Tables

**Figure 1 jcm-12-04958-f001:**
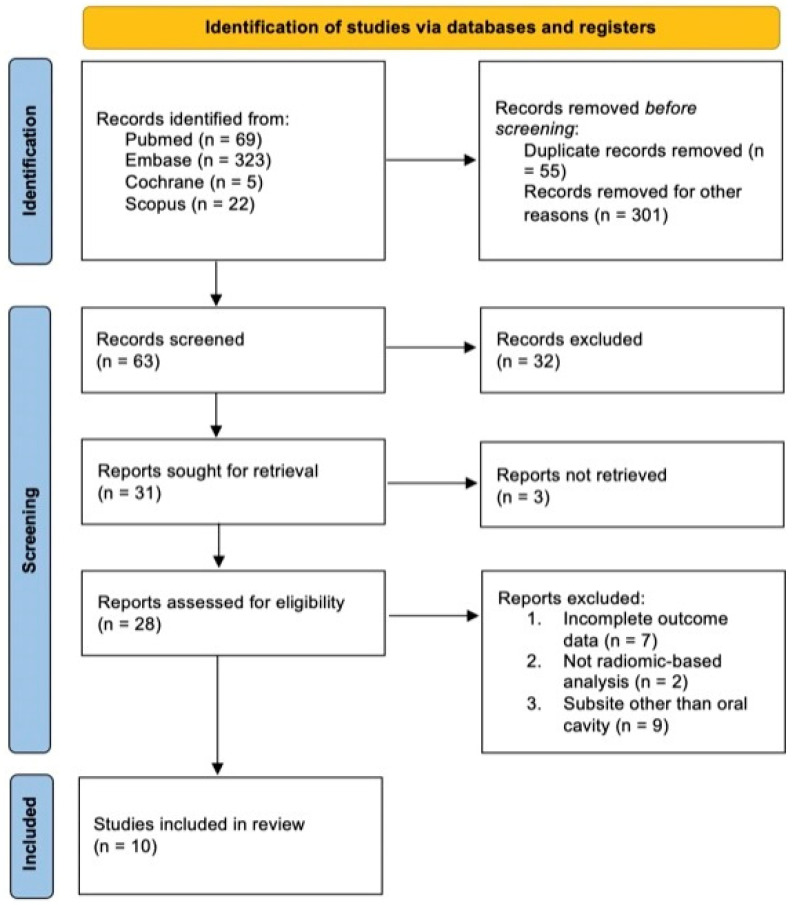
PRISMA flow diagram.

**Figure 2 jcm-12-04958-f002:**
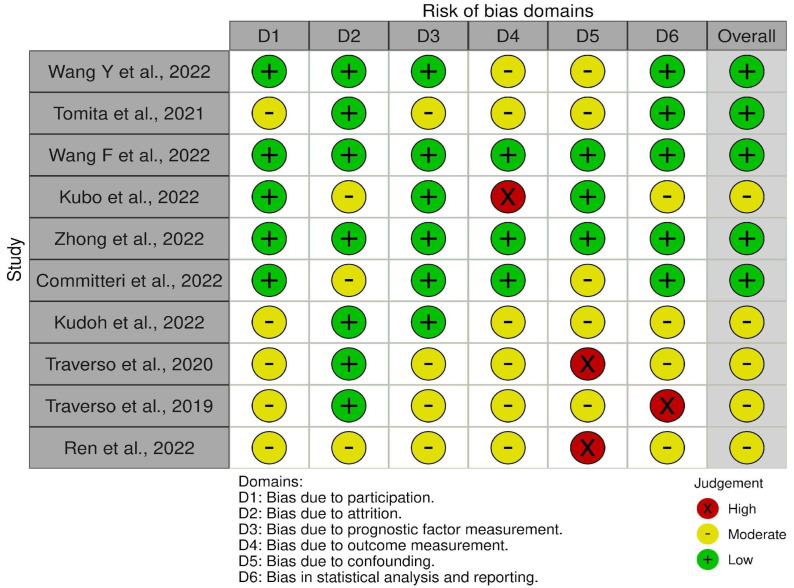
Traffic lights plot [24,25,26,27,28,29,30,31,32,33].

**Table 1 jcm-12-04958-t001:** Characteristics of the included studies. MRI, magnetic resonance imaging; CT, computed tomography; PET, positron emission tomography; LN, lymph node; NA, not available; pts, patients.

Study	Country	Study Type (Retrospective R, Prospective P)	Imaging Technique (MRI 1, CT 2, PET 3)	Feature Extraction	Software	Year of Recruitment	Sample Size (*n*)	Primary/Train Cohort (%)	Validation/Test Cohort (%)	Subsite (%)	Staging	Positive LNs (*n*)	Negative LNs (*n*)
Wang Y et al., 2022 [24]	China	R	1	LN	LIFEx (EVOMICS)	2013– 2021	160	75	25	100 oral cavity (70.1 tongue)	61 I–II 39 III–IV	NA	NA
Tomita et al., 2021 [25]	Japan	R	2	LN	Python	2013– 2017	44	70	30	100 oral (tongue, gingiva, floor of mouth)	I–IV	51	150
Wang F et al., 2022 [26]	China	R	1	Tumor	Python (version 3.5.2)	2012– 2019	236	67	33	100 tongue	I–IV	99	137
Kubo et al., 2022 [27]	Japan	R	2	LN	Python (Pyradiomics software)	2008– 2019	161	NA	NA	100 tongue	I–III	63	NA
Zhong et al., 2022 [28]	China	R	2	Tumor	Matlab 2018b (MathWork)	2013– 2018	313	60	40	100 tongue	I–IV	143	170
Committeri et al., 2022 [29]	Italy	R	2	Tumor	PyRadiomics	2016– 2020	81	80	20	100 tongue	I–II	NA	NA
Kudoh et al., 2022 [30]	Japan	R	3	Tumor	Matlab	2015– 2019	40	80	20	100 tongue	15 I, 30 II, 18 III, 37 IV	19 pts	21 pts
Traverso et al., 2020 [31]	Multicentric	R	1	NA	PyRadiomics v2.1.2	2003– 2017	243	70	30	100 oral	NA	NA	NA
Traverso et al., 2019 [32]	Multicentric	R	1	Tumor	PyRadiomics	NA	134	80	20	100 oral	NA	NA	NA
Ren et al., 2022 [33]	China	R	1	Tumor	Pyradiomics	2015– 2021	55	NA	NA	100 tongue	I–II	21 pts	34 pts

**Table 2 jcm-12-04958-t002:** Diagnostic performance of the included studies. ACC, accuracy; AUC, Area Under the Curve; NA, not available.

Study	Sensitivity	Specificity	ACC (95%CI)	AUC (95%CI)
Wang Y et al., 2022 [24]	0.85	0.71	0.79	0.82
Tomita et al., 2021 [25]	0.74	0.88	0.85	0.85
Wang F et al., 2022 [26]	0.95	0.98	0.97	0.99
Kubo et al., 2022 [27]	NA	NA	0.85	0.92
Zhong et al., 2022 [28]	0.82	0.87	0.84	0.91
Committeri et al., 2022 [29]	0.94	0.98	0.96	0.93
Kudoh et al., 2022 [30]	0.65	0.70	0.68 ± 0.13	0.79
Traverso et al., 2020 [31]	NA	NA	0.70 (0.67–0.71)	NA
Traverso et al., 2019 [32]	NA	NA	NA	0.83
Ren et al., 2022 [33]	0.79	0.86	0.82	0.87 (0.77–0.96)

## Data Availability

The data that support the findings of this study are openly available in Pubmed, Embase, Scholar.

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
