# Peer review of "Radiomics-Based Analysis in the Prediction of Occult Lymph Node Metastases in Patients with Oral Cancer: A Systematic Review"

_jcm, 2023, doi:10.3390/jcm12154958_

Round 1

Reviewer 1 Report

It is an interesting manuscript because it provides indications that when radiomics are being integrated with clinicopathological factors, this may result in improved diagnosis or prediction. However, their methodology to answer the primary issue (prediction of occult nodal metastasis-ONM) is not clear to me. More specifically, I did not understand what diagnostic quantitative data (sensitivity, auc etc) in articles examining tumour and not lymph nodes represents and in what way this is related to prediction of occult lymph node metastases since in some of these studies this data was not available.

Author Response

I Reviewer

It is an interesting manuscript because it provides indications that when radiomics are being integrated with clinicopathological factors, this may result in improved diagnosis or prediction. However, their methodology to answer the primary issue (prediction of occult nodal metastasis-ONM) is not clear to me. More specifically, I did not understand what diagnostic quantitative data (sensitivity, auc etc) in articles examining tumour and not lymph nodes represents and in what way this is related to prediction of occult lymph node metastases since in some of these studies this data was not available.

Dear Reviewer, thanks for your interesting remarks and forgive us for being unclear. Indeed, a clear-cut distinction must be made between features extracted from the tumor in radiologically cN0 necks and features extracted from the lymph nodes themselves. Since only three articles used nodal levels in this regard, we have discussed in detail the analytical methods used by the authors and the clinical relevance of their findings. We hope to have made our points clearer in our revised manuscript.

Reviewer 2 Report

The authors conducted a systematic review to define the role of radiomics-based analysis in the prediction of lymph node status in patients with oral cancer.

The topic is interesting and relatively novel.

The manuscript has been well written and discussed.

Minor comment:

Why did the authors not include "oral carcinoma" in the keywords of the study?

Author Response

Reviewer II

The authors conducted a systematic review to define the role of radiomics-based analysis in the prediction of lymph node status in patients with oral cancer.

The topic is interesting and relatively novel.

The manuscript has been well written and discussed.

Minor comment:

Why did the authors not include "oral carcinoma" in the keywords of the study?

Dear Reviewer, thank you very much for your favorable comments on our paper. We have amended the keywords as per your request.

Reviewer 3 Report

The authors Jiang et al. present a literature review about the use of radiomics analyses in oral cavity squamous cell carcinoma (OSCC). In their search, they identified ten studies that they have included in their analysis and discussed. They conclude that the addition of ML-based models in the assessment of preoperative imaging may improve the detection of pathological lymph nodes in OSCC patients though a validation study is needed.

Although the topic is not new, the presented manuscript is a solid and balanced review on lymph node prediction in OSCC/HNSCC.

One point that is only briefly discussed is that most of the identified studies included only the tumor site as regions of interest for the radiomics analyses. For example, Haider et al. had tried both tumor, lymph node and both as regions of interest in oropharyngeal SCC and PET CT radiomics. It seems that tumor radiomics are more useful because in radiomic analyses it can be hard to include all or the relevant metastatic lymph nodes as the ROI, especially in cN0 staged neck imaging and questionable occult LN metastases. I would suggest to discuss this point a bit more thoroughly.

English language is fine.

Author Response

Reviewer III

The authors Jiang et al. present a literature review about the use of radiomics analyses in oral cavity squamous cell carcinoma (OSCC). In their search, they identified ten studies that they have included in their analysis and discussed. They conclude that the addition of ML-based models in the assessment of preoperative imaging may improve the detection of pathological lymph nodes in OSCC patients though a validation study is needed.

Although the topic is not new, the presented manuscript is a solid and balanced review on lymph node prediction in OSCC/HNSCC.

One point that is only briefly discussed is that most of the identified studies included only the tumor site as regions of interest for the radiomics analyses. For example, Haider et al. had tried both tumor, lymph node and both as regions of interest in oropharyngeal SCC and PET CT radiomics. It seems that tumor radiomics are more useful because in radiomic analyses it can be hard to include all or the relevant metastatic lymph nodes as the ROI, especially in cN0 staged neck imaging and questionable occult LN metastases. I would suggest to discuss this point a bit more thoroughly.

English language is fine.

Dear Reviewer, thank you for your precious remarks. As also pointed out by the reviewer number 1, in the first version of our manuscript we have not adequately stressed the two distinct situations where ML was applied. In seven out of the ten articles analyzed, features used for building the ML-based model were extracted from the tumor in radiologically cN0 necks, while in the remaining three radiological features were taken from ROI set on the lymph nodes themselves. Because each situation yields profoundly different implications (ie., if radiologically suspicious lymph nodes are already present, the value of ML in predicting “occult” metastases simply does make any sense), we have decided to discuss in detail the analytical methods used by the authors of the last three articles. We hope to have made our text clearer in our revised manuscript.
